# Validation of global precipitation time series products against tree ring records and remotely sensed vegetation greenness

**Vinicius Manvailer**[1,2]*, **Andreas Hamann**[1]

**1** Department of Renewable Resources, University of Alberta, Edmonton, AB, Canada, **2** Department of Natural Resources Canada, Canadian Forest Service, Victoria, BC, Canada

* vinicius.manvailergoncalves@nrcan-rncan.gc.ca

**Data Availability Statement:** All data used in this publication are already publicly available, and cited accordingly. We also deposited our code and intermediate data products (e.g. extracted and summarized remote sensing data) in an open

## Abstract

Global interpolated climate products are widely used in ecological research to investigate biosphere-climate interactions and to track ecological response to climate variability and climate change. In turn, biological data could also be used for an independent validation of one aspect of climate data quality. All else being equal, more variance explained in biological data identifies the better climate data product. Here, we compare seven global precipitation time series products, including gauge-based datasets (CRU-TS, UDEL-TS, GPCC), re-analysis products (ERA5, CHELSA), a satellite-based dataset (PERSIANN) and a multi-source product that draws on gauge, re-analysis, and satellite data (MSWEP). We focus on precipitation variables, because they are more difficult to interpolate than temperature, and show larger divergence among gridded data products. Our validation is based on 20 years of remotely sensed vegetation greenness (MODIS-EVI) and 120 years of tree ring records from the International Tree Ring Data Bank (ITRDB). The results for the 20-year EVI based validation shows that all gauge and re-analysis data products performed similarly, but were outperformed by the multi-source MSWEP product, especially in regions with low weather station coverage, such as Africa. For analyzing long 120-year time-series, UDEL-TS showed superior performance prior to the 1940s, with especially large margins for northern Asia and the Himalayas region. For other regions, CRU-TS and GPCC could be recommended. We provide maps that can guide the best regional choice of climate product for research involving time series of biological response to historic climate variability and climate change.

## Introduction

Researching biological response to interannual climate variability, long-term climate trends or climate extreme events requires reliable historical climate data. Such data are usually provided as gridded data products that have been interpolated from weather stations, allowing for estimates of climate variables for any study site or sample location [1–3]. However, depending on the climate variables required, the topographic complexity of the landscape, and the distance

access repository. The link has been included in the manuscript and in the cover letter: https://doi.org/10.6084/m9.figshare.24540928.

**Funding:** A.H. RGPIN-330527-20 Natural Sciences and Engineering Research Council of Canada (NSERC) https://www.nserc-crsng.gc.ca The funders had no role in study design, data collection and analysis, decision to publish, or preparation of the manuscript.

**Competing interests:** The authors have declared that no competing interests exist.

between the study site of interest and the nearest weather station, climate estimates from different data products may be very variable in quality [4–6]. Generally, temperature variables are easier to interpolate and even in mountainous areas they follow predictable patterns according to adiabatic or environmental lapse rates. This is not the case for precipitation variables that are driven by more difficult to model processes, such as orographic lift and rain shadows that do not scale in straight-forward ways with elevation. Furthermore, the shorter the historical time period to be predicted (annual, monthly or daily), the more precipitation estimates are driven by spatially distinctly bounded, and randomly occurring weather events, making interpolations especially challenging [7].

Nevertheless, at monthly spatial resolution, global and regional gridded climate data products are available from many sources, and these products are based on two general types of data sources. Traditionally, gridded climate data is produced from interpolating weather station data (gauge-base) using a variety of interpolation methods [8,9]. Since approximately the year 2000, time series climate data products based on remotely sensed temperature and precipitation have become available from high-quality satellite-based sensors [10]. However, both approaches have their own limitations. Gauge-base data provides long-term climate datasets (100+ years) but is often severely limited by regional and temporal gaps in weather station coverage [11]. Remote sensing-based datasets, on the other hand, have homogeneous coverage of the earth's surface that are covered by the orbital path of the satellites (between 60° latitude north and south), but the earliest datasets only start around the 1980s with high quality data only available since around the year 2000. Some gridded climate data products make use of both data sources, where remotely sensed climate estimates are used where available, and adjusted with weather station information [12].

Making selections among different climate data products could therefore potentially be an important choice for researchers studying climate-biosphere interactions. For gauge-based precipitation products, there are two main reasons why estimates may differ in precision and accuracy. First, authors of different interpolated climate data products may have different levels of access to weather station data from specific countries or regions [13], and second, different interpolation algorithms used may result in varying regional quality of climate variable estimates [1,13]. The choice of interpolation algorithms is not particularly important for regions with dense weather station coverage, where most methods yield very similar estimates and validation statistics. However, for remote regions with sparse or no weather station coverage, such as high montane, high latitude, or other undeveloped regions of the world, interpolation methods can substantially diverge in their climate estimates [13,14]. This divergence among different interpolation approaches often goes undetected because areas without training data also have no weather stations available for validation.

While validation of the accuracy of climate data is usually performed against weather station data by the authors of climate data products, these validations are generally meaningless for cross-product comparisons. First, there are many valid approaches to validation including various statistical methods, different options to subset training and validation data, and different approaches to deal with autocorrelations among nearby weather station records that may inflate validation statistics [15–18]. As such, validation statistics from different studies are not directly comparable. Second, after validation statistics have been computed, the withheld validation data will be re-joined with the training data to create the final climate data product to take advantage of all available information. For valid product comparisons, all authors would need to use the same validation data, withheld from the training data, and as such *post hoc* product comparisons with weather station records are not possible.

Biological data, on the other hand, offers an independent data source for validating one aspect of the quality of climate data products. In this study we use 20 years of global historical

remote sensing records of vegetation greenness (MODIS-EVI) and 120 years of tree ring records from the International Tree Ring Data Bank (ITRDB). This independent validation is based on the strength of plant-climate associations, with the expectation that variance explained in biological data should be higher for better quality climate data sets. We note that this is only a *partial* validation, because the accuracy of absolute climate values cannot be assessed with biological data sources. However, we can assess the precision of plant-climate interactions in time series. To describe this limitation in other words, we cannot assess systematic bias of climate estimates for any location with biological data, but statistical precision of climate data tracking inter-annual variation of historical biological records can be quantified. For this comparison, we can hold all model parameters and top-level attributes (such as missing values and length of data coverage) constant, with only the climate data source varying in the validation analysis.

This study, contributes such an independent validation effort for seven widely used global historical climate products CRU v4, UDEL-TS, GPCC, ERA5, CHELSA, MSWEP and PER-SIANN, described in more detail below. Our objective is to carry out two types of comparisons: one long-term evaluation against 120 years of tree ring records for three weather station-derived products with the same long time series coverage (CRU v4, UDEL-TS, GPCC). In a second comparison, we use 20 years of recent global historical remote sensing records of vegetation greenness for validation. For more recent time periods, remote sensing based products (PERSIANN), re-analysis products that combine observational data with general circulation models (ERA5, CHELSA), and multi-source models that combine gauge, satellite, and re-analysis data (MSWEP) could yield better precipitation estimates, especially for regions with sparse weather station coverage. Our objective is to provide region-specific guidance to researchers, who investigate time series of biological response to historic climate variability and climate change, as to which climate product is most suitable for their research.

## Methods

### Climate data products

We selected seven widely used global, monthly climate data products with a historical coverage dating to the beginning of the 21$^{st}$ century for gridded products derived from weather station records, and dating back to the 1980s for remote-sensing or reanalysis based climate products. This includes the CRU TS 4.05 dataset from the Climatic Research Unit of the University of East Anglia [8], the Terrestrial Precipitation product from the University of Delaware (UDEL) [19], the Full Data Monthly Product v2018 from the Global Precipitation Climatology Centre (GPCC) [20], the 5th generation reanalysis product (ERA5) of the European Centre for Medium-Range Weather Forecasts [12], the climatologies at high resolution for the earth's land surface areas v2.1 (CHELSA) from the Swiss Federal Institute for Forest, Snow and Landscape Research [21], the Precipitation Estimation from Remotely Sensed Information using Artificial Neural Networks—Climate Data Record (PERSIANN-CDR) from the Center for Hydrometeorology and Remote Sensing at the University of California [10], and the Multi-Source Weighted-Ensemble Precipitation (MSWEP) product from Department of Civil and Environmental Engineering at the Princeton University [22]. Attributes of these climate data products are summarized in Table 1. A general map of weather stations with at least 30 years of precipitation data is shown in Fig 1, based on Castellanos and Hamann [11], to represent general regional climate data coverage. However, not all of the above gauge-based climate products would have utilized all of these stations for interpolation, depending on different exclusion criteria.

**Table 1. Global interpolated precipitation products evaluated in this study.** Datasets were generated by the University of East Anglia Climatic Research Unit (CRU), the University of Delaware Terrestrial Precipitation (UDEL), the Global Precipitation Climatology Centre (GPCC), the European Centre for Medium-Range Weather Forecasts Reanalysis v5 (ERA5), the Swiss Federal Institute for Forest, Snow and Landscape Research (CHELSA), the Center for Hydrometeorology and Remote Sensing at the University of California (PERSIANN), and the Department of Civil and Environmental Engineering at Princeton University (MSWEP).

| Dataset | Type | Resolution | Start | End[1] |
|---|---|---|---|---|
| CRU4 | Gauge | 0.5˚ | 1901 | 2022 |
| UDEL | Gauge | 0.5˚ | 1901 | 2017 |
| GPCC | Gauge | 0.25˚ | 1890 | 2022 |
| ERA5 | Reanalysis | 0.25˚ | 1950 | 2022 |
| CHELSA | Reanalysis | 30" | 1979 | 2022 |
| PERSIANN | Satellite | 0.25˚ | 1983 | 2022 |
| MSWEP | Multi-source | 0.01˚ | 1979 | 2022 |

[1]) at the time of data access, 2022 implies ongoing updates.

## Tree ring records

A global dataset of tree ring records was obtained from the International Tree Ring Data Bank (ITRDB) [23,24]. Raw tree ring measurements from approximately 150,000 cores collected at 4422 sites were used to develop site- and species-specific chronologies. Chronologies were detrended using the Friedman Super-Smoother method to remove both long- and medium-frequency variability and maintain short-frequency (year-to-year) variability. This removes age-related trends in tree ring data that may be confounded with long-term climate trends or decadal climate oscillations. Our analysis therefore evaluates how well tree ring data tracks inter-annual climate variability. The Friedman Super-Smoother method was implemented with the package dplR [25–27] for the R programming environment [28]. We used the default parameters of the function *detrend()*, which were equal case weights (*wt* parameter), automatic

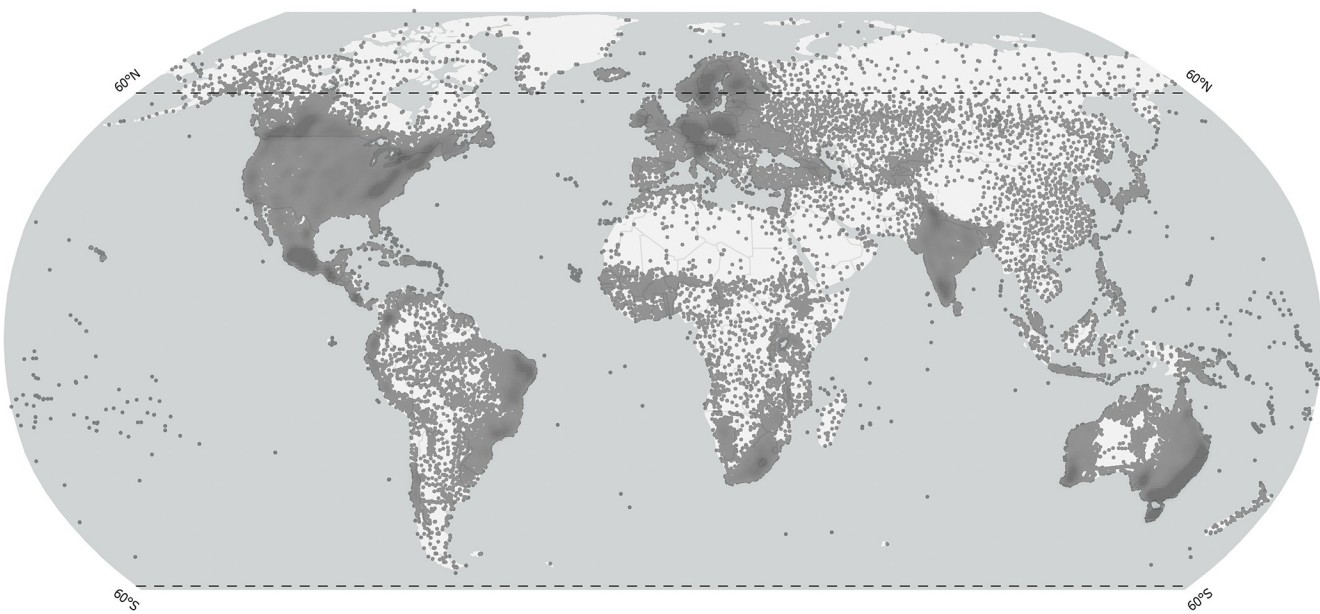

**Fig 1. Spatial coverage of weather stations with precipitation records.** Remotely sensed precipitation estimates are not available above 60˚ latitude, indicated by the dashed line. The figure uses public domain spatial data from Natural Earth (http://www.naturalearthdata.com/) and public domain location data from the International Tree Ring Databank (https://www.ncei.noaa.gov/products/paleoclimatology/tree-ring).

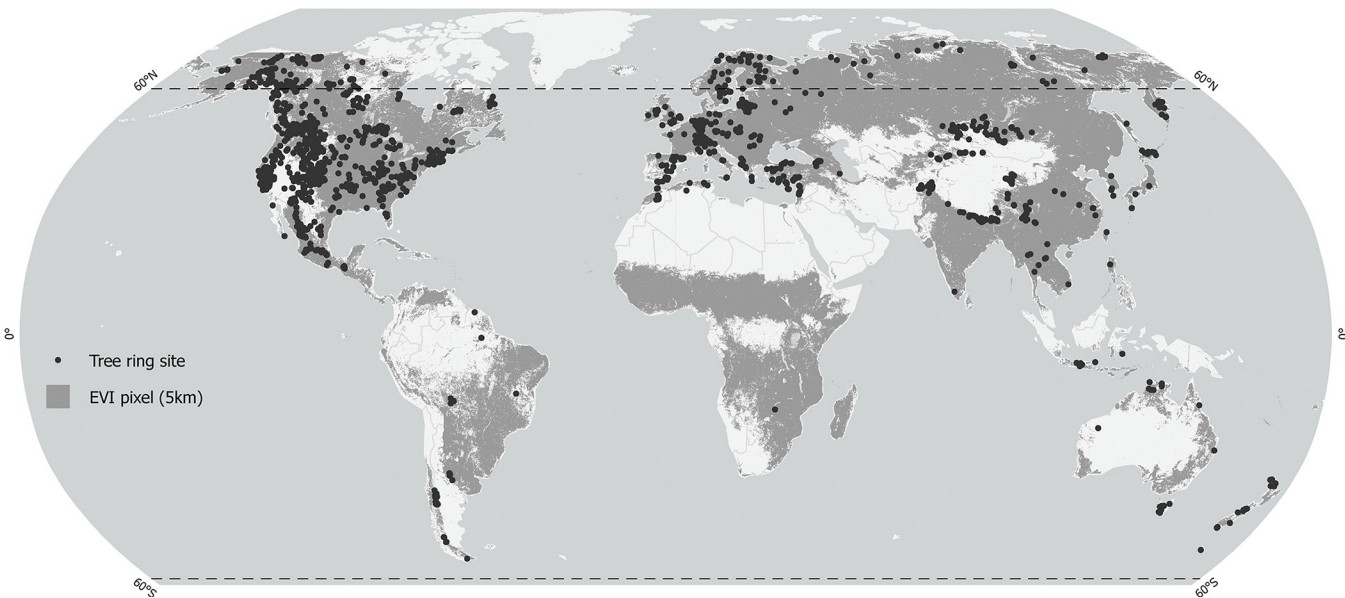

**Fig 2. Spatial coverage of tree ring chronologies and remotely sensed vegetation greenness.** Remotely sensed enhanced vegetation index (EVI) coverage is restricted to pixels with a dominant growing season, allowing for an annual area under the curve estimate from EVI data as a proxy for vegetation productivity, equivalent to tree ring widths. The figure uses public domain spatial data from Natural Earth (http://www.naturalearthdata.com/) and an original spatial layer developed from open-access EVI2 data (https://lpdaac.usgs.gov/products/mcd12q2v006/).

span detection using cross-validation (*span* = "cv") and default bass (*bass* = 0). After detrending, each series of cross-dated ring width measurements from the same site and of the same species were averaged using a robust biweighted mean to reduce the effect of outliers in the final chronology, implemented with the function *chron()*, of the package dplR [25–27] for the R programming environment [28]. The function *chron()*, was also used to fit an autoregressive model to remove temporal autocorrelation (pre-whitening), which maximized correlations with precipitation data (determined empirically, post hoc). The spatial coverage of tree ring records is shown in Fig 2, and the overlap of chronology records with climate data products is shown in Fig 3 The validation was restricted to complete pairwise comparisons across all data products. In other words, all data products were evaluated based on an identical number of chronologies and years per chronology.

## Remote sensing data

The most widely used vegetation indices to assess vegetation greenness and draw inferences on vegetation health and productivity are the Normalized Difference Vegetation Index (NDVI) and the Enhanced Vegetation Index (EVI) [29]. While both have been used widely, EVI is less prone to noise from atmospheric and soil conditions and less easily saturated by dense vegetation, and therefore more sensitive to interannual variation of forested areas, particularly in temperate and tropical regions [30,31]. Here we use an annual EVI data from the NASA Earth Observing System Data and Information System (EOSDIS) website, where an annual area under the curve has been integrated (collection MCD12Q2 v006, product MOD13A1), which represents photosynthesis over the entire growth cycle from greenup to dormancy [32]. This data is similar in principle to tree-ring records, where the ring width represents the integration of a growing season's growth. Also similar to tree ring data the EVI area under the curve records are only available for climate regions with a distinct growing seasons (Fig 2, shown as dark gray data coverage). We aggregated the original 500 m resolution EVI

area under the curve data to 5 km resolution and included only pixels that had time series with at least 50% non-missing values for a 5 km aggregated grid cell. Our analysis focuses on forested areas, by excluding croplands, grasslands and other non-forested areas using the vegetation continuous fields product (MOD44B) [33]. This aggregated 5 km resolution dataset is available through a data repository (http://doi.org/ 10.6084/m9.figshare.24540928), and could be used to test other climate products.

## Statistical analyses

For each tree ring site and for each pixel of the EVI dataset we carry out an 8-months lagged correlation analysis. We assumed the growing season to have ended at the end of August for the northern hemisphere and at the end of February for the southern hemisphere, evaluating cumulative annual EVI or tree ring width with 8 months of climate data with a simple linear model, implemented with the *lm()* function of the R base package [28]. Although more sophisticated multivariate response function analysis methods are available for research in dendroclimatology, this is not needed in this study. For comparisons of climate data products with all factors held identical, a simple statistic from a correlative model is sufficient. We therefore chose an un-adjusted variance explained ($R^2$) from a multiple regression model, where the EVI area under the curve or tree ring width was specified as response variable, and the 8 month of monthly climate data prior to the end of the growing season were the predictor variables. This generally captures the growing season and several month prior where precipitation-growth correlations are expected to be reasonably strong.

To concisely report results for thousands of tree ring chronologies and millions of EVI grid cells, we used regional aggregation of the resulting $R^2$ values. EVI data was aggregated using ecozone delineations from the Terrestrial Ecoregions of the World product [34]. To enable regional choices of the best climate data product, an assessment of the best dataset for an ecoregion was made using the Condorcet voting algorithm. The Condorcet method is a ranked pair-wise balloting system. In our case, the ranked ballots are the ranked $R^2$ statistics for each tree ring chronology or EVI pixel that belong to the same ecoregion. A ranked ballot not only evaluates how often a product is ranked first in regional summaries. Last place rankings (and intermediate rankings) carry negative (or neutral) weight in the evaluation as well. The ranked ballot procedure was implemented with the *Condorcet()* function of the *Vote* package (Version 2.3–2) [35] for the R programming environment [28].

Lastly, for the long-term historical analysis, using the 120-year dendrochronology dataset, we used the same multiple regression approach described above using 21-year moving windows in single year increment steps, covering the period 1901 to 2000, and therefore reporting moving average statistics from 1911 to 1990. This allows the evaluation of how data quality has developed over time. All graphical representation of quantitative data were generated with the *ggplot2* package [36] for the R programming environment, and maps of data coverage and results were generated with ArcGIS Pro v3.0.0 [37].

## Results & discussion

### Regional comparisons based on EVI data

Remote-sensing based validation statistics of precipitation products for recent climate data since 2000 are very similar, when summarized by continent or subcontinent (Fig 4, bar charts). It should be noted that the analysis is restricted to forested areas with seasonal growth patterns and therefore excludes ecoregions with aseasonal tropical rainforests and areas with minimal tree coverage. Regional Condorcet ranked-ballot winners by ecoregions provides spatially more explicit results (Fig 4, map). All data products appear as the best choice in some regions

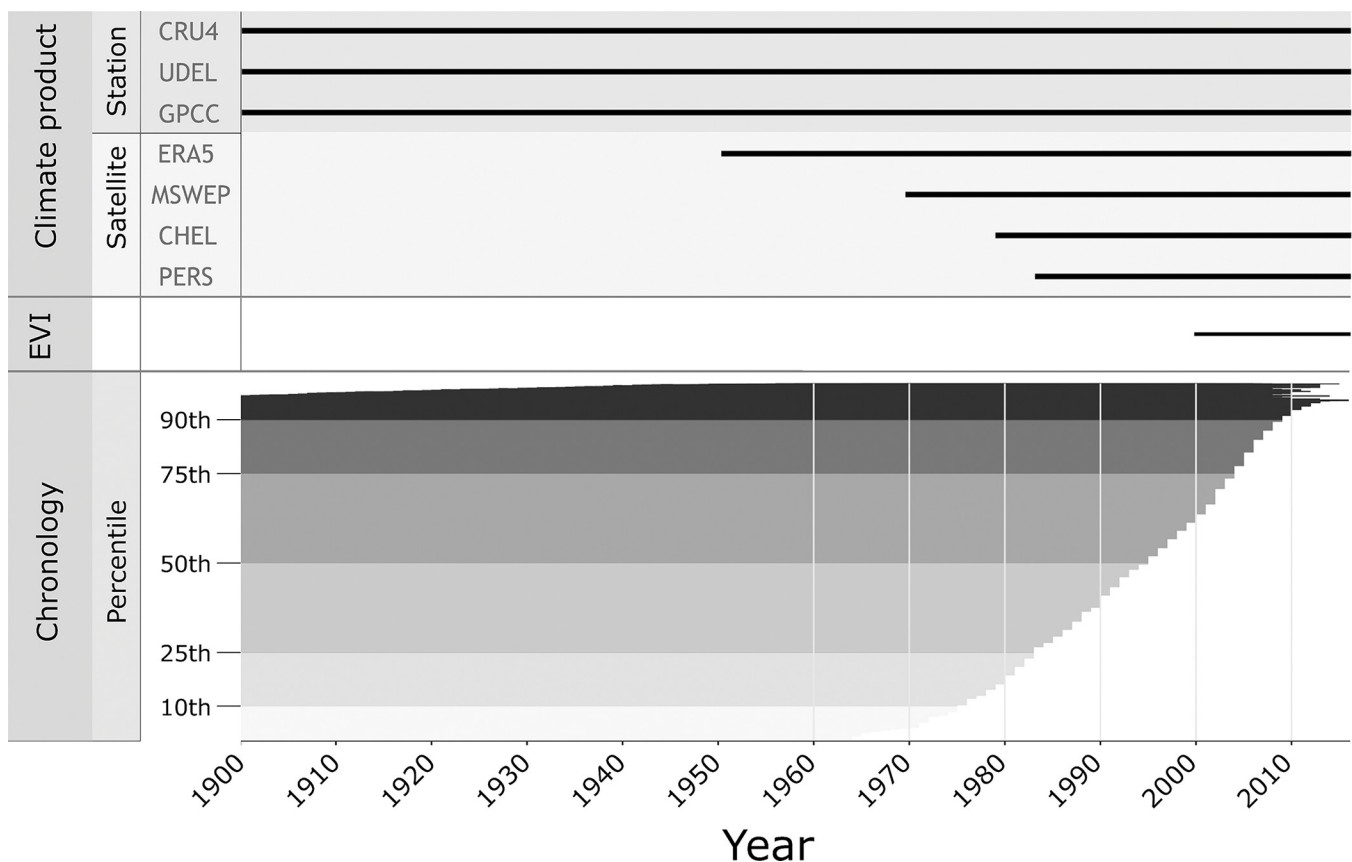

**Fig 3. Temporal data coverage of climate products, remotely sensed EVI data, and tree ring chronologies.** Tree ring chronologies are ordered by end year and truncated at 1900s to match climate data (total of 4422 sites). Shades of grey in lower panel represent the percentiles of the dataset. Only temporally pairwise-complete data was used for climate product comparisons.

of the world, notably, however, MSWEP is more frequently the winner dominating large area of several ecosystems. For North America, MSWEP, CHELSA and UDEL dominate as the preferred data products by a small margin. Northern Asia is dominated by MSWEP and CRU4. For sub-Saharan Africa and the Sierras of Mexico, MSWEP wins by relatively large margin when compared to all others.

A fairly consistent result is that the remotely-sensed PERSIANN precipitation product under-performs in most regions of the world (most frequently ranked last), although the product does come out as ranked-ballot winner for mid- to high-latitude eastern continental regions, i.e. the east coast of Canada, the US, Russia and China, as well as similarly positioned mid-latitude eastern regions of South America and the Africa on the southern hemisphere. Climatic patterns, typically with year-round intermediate amounts of rainfall in these regions appear to favor accurate estimates of precipitation via remote sensing. We speculate that both passive and active microwave sensors may be less likely to be saturated, and therefore allow for better precision in regions with intermediate rainfall. While the remote-sensing based PERSIANN product was not superior to gauge-based products where weather station coverage is sparse, a multi-source models that combines gauge, satellite, and re-analysis data (MSWEP) could yield better precipitation estimates. While superiority of MSWEP compared to other products was not very closely linked to poor weather station coverage (c.f., Figs 1 and 4), impressive margins of improvement were observed for some regions with few ground-based climate stations, such as Africa and parts of South America

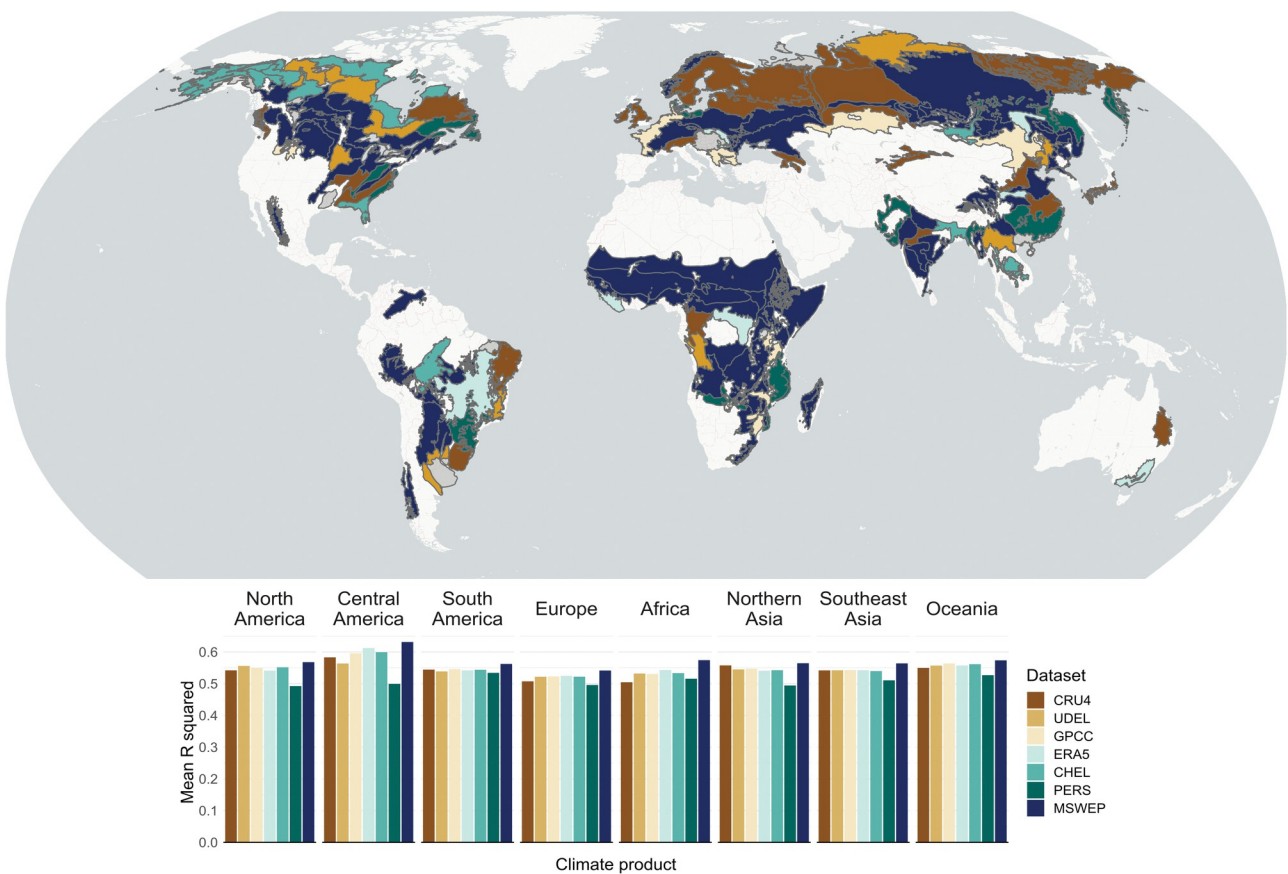

**Fig 4. Best regional interpolated precipitation products for a recent 2000–2017 period against remotely sensed vegetation greenness.** The comparisons are based on the strongest correlation with Enhanced Vegetation Index (EVI) annual area under the curve values. The map by ecoregions represents the best performing precipitation product, using the Condorcet winner method, where $R^2$ values of individual EVI pixels are used equivalently to ranked ballots. The figure uses public domain data from Natural Earth (http://www.naturalearthdata.com/) and original results generated in this study.

## Validation against long-term tree ring data

While validation against EVI time series provides spatially explicit results, tree ring records can evaluate quality of data products over long time periods. For this second comparison, only gauge-based products (CRU, UDEL and GPCC) can be included, where data coverage is available since the beginning of the century.

Variance explained by moving averages of 21-year windows reveal that the UDEL is generally superior for early climate estimates based on data from 1901–1940, corresponding to $R^2$ values of moving windows from 1910–1930 (Fig 5, global panel). Subsequently, $R^2$ values are fairly similar for moving window mid-points from 1930 to 1960, and then separate by a moderate amount for the most recent decades, with GPCC performing best, followed by UDEL and CRU4. It is important to note that the superior performance of UDEL in the early decades only applies to specific regions, namely Northern Asia and the Himalayas region, where whether station coverage is generally sparse, and we speculate that the researchers developing the UDEL product may have had access to station records not available for interpolation in CRU and GPCC products. However, UDEL data is also marginally better in early century climate estimates for North America, Europe, South America and Southeast Asia. Their method of interpolation, an angular-distance weighting (ADW) method, appears superior to those

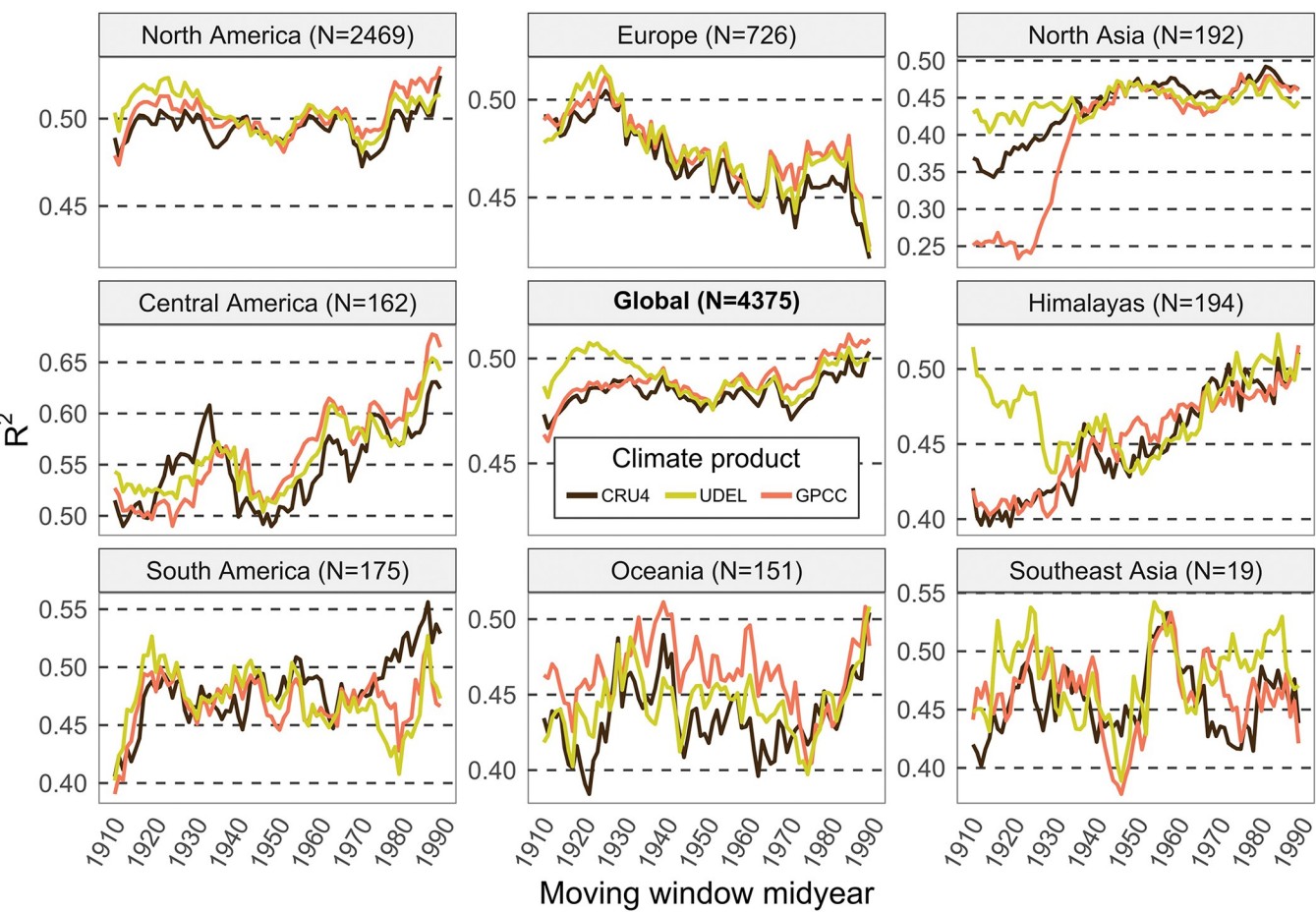

**Fig 5. Best regional interpolated precipitation products in a long term (1901–2017) evaluation against tree ring records.** Trends over time in validation statistics for the three global interpolated precipitation products that extend back to the 1900. The lines represent variance explained in tree ring with by precipitation using 20-year moving windows.

used in other products for early century reconstructions, where weather station coverage is generally sparse. AWD can make use of directional information from data nodes, and has been found to handle sparse data coverage reasonably well [38–40].

Regional breakdowns suggest that all three gauge-based climate data products perform fairly similarly besides regional differences in the early decades (Fig 5, regional panels). That said, a small but notable difference among data products is CRU outperforming other data products for recent climate since the 1970s for South America. For the Oceania region GPCC appears to be the best choice for most of the evaluated time series.

As a side note, not directly related to this study's objectives, we should also briefly interpret trends and variability in validation statistics over time that are visible in all data products. A general increase in the strength of the association between precipitation variables and chronology time series indicates that biological response has become more dependent on variability in precipitation over time. This is apparent for Northern Asia, the Himalayas region, and Central America. An opposite trend can be observed for Europe, indicating a weakening of the dependence of tree growth on variability in precipitation. Directional climate change towards drier conditions and/or warmer temperatures would be a plausible explanation for positive trends, and this has in fact been observed, especially for Central America (with a strong trend toward less precipitation [41–43], and the Himalayas with a strong warming trend of high elevation

ecosystems where trees were sampled [44–46]. The weakening in of plant-climate interactions in Europe can be explained by increased water-use efficiency from cumulative $NO_x$ pollution due to industrialization over the century, where increases in water-use efficiency can over-compensate for any negative climate change effects on tree growth (e.g., Guerrieri [47]).

## Limitations of the analysis

As noted previously, our contribution is only a *partial* validation of climate data products, because the accuracy of absolute climate values cannot be assessed with biological data sources. Our evaluation is therefore only relevant for a specific target audience of researchers who analyze historical time series in biological or geophysical data. To establish a putatively causal link between climate and biological response, with correlative or associative methods using time series data, bias is not an important metric. Climate data can actually be extremely biased without compromising such analyses. To give an example, consider the usage of coarse resolution gridded temperature data used in mountainous terrain. Widely used datasets for this purpose such as CRU or UDEL have 0.5˚ resolution, and absolute temperature values along elevation gradients can vary extremely across an approximately 50×50 km grid cell in mountainous terrain. Yet, in the field of dendroclimatology this poses no problem, because a warmer than average growing season in a particular year is warmer than average at all elevations. That said, our evaluation should not inform data selection for types of analysis that require unbiased estimates. To give an example, analysis types that build on climatic comparisons among different locations for a fixed time period, such as ecological niche modeling or species distribution modeling should not rely on climate data recommendations from this analysis.

## Conclusions

For researchers who analyze historical time series in biological or geophysical data, our analysis allows to infer the best regional choice of time-series precipitation products. That said we should emphasize that for applications that require unbiased estimate of differences among locations for a fixed time period, our recommendations are not applicable. The results for the 20-year time series of remotely sensed EVI allows for comprehensive regional comparisons. All gauge and re-analysis based data products performed similarly, but they were outperformed by the multi-source MSWEP product that utilizes gauge, re-analysis, and remote sensing data. Especially for regions with low weather station coverage, such as Africa and parts of South America, the margin of improvement in how well MSWEP time series data tracks vegetation greenness was impressive. The analysis confirmed our working hypothesis that satellite-based or multi-source products have the potential to provide better precipitation estimates in regions of low weather station coverage. Unfortunately, satellite or re-analysis based products are not available prior to the 1980s. For applications that require longer time series analysis, gauge based products are the only option. Of the three gauge-based data products included in the analysis against 120-year tree ring records (CRU, GPCC and UDEL), the results showed that UDEL had superior performance prior to the 1940s, with especially large margins for northern Asia and the Himalayas region. From the 1940s onward, all three gauge-based climate data products perform fairly similarly. Locally, CRU outperformed other data products after the 1960s for South America. For the Oceania region GPCC was the best choice for analysis of long time series.

## Acknowledgments

We thank the custodians and all contributing authors of the International Tree Ring Database for creating this valuable resource.

## Author Contributions

**Conceptualization:** Vinicius Manvailer, Andreas Hamann.

**Data curation:** Vinicius Manvailer.

**Formal analysis:** Vinicius Manvailer.

**Funding acquisition:** Andreas Hamann.

**Investigation:** Vinicius Manvailer.

**Methodology:** Vinicius Manvailer, Andreas Hamann.

**Resources:** Andreas Hamann.

**Supervision:** Andreas Hamann.

**Validation:** Vinicius Manvailer.

**Visualization:** Vinicius Manvailer.

**Writing – original draft:** Vinicius Manvailer.

**Writing – review & editing:** Andreas Hamann.

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
