## [Decision Letter · Decision Letter 0]

5 Jun 2023

PONE-D-23-14494Validation of global precipitation time series products against tree ring records and remotely sensed vegetation greennessPLOS ONE

Dear Dr. Manvailer,

Thank you for submitting your manuscript to PLOS ONE. After careful consideration, we feel that it has merit but does not fully meet PLOS ONE’s publication criteria as it currently stands. Therefore, we invite you to submit a revised version of the manuscript that addresses the points raised during the review process.

We look forward to receiving your revised manuscript.

Kind regards,

Nir Y. Krakauer

Academic Editor

PLOS ONE

Journal Requirements:

   "We thank the custodians and all contributing authors of the International Tree Ring Database for creating this valuable resource. Funding for this study was provided by the NSERC Discovery Grant RGPIN-330527-20 to AH and a Coordenadoria de Aperfeiçoamento de Pessoal de Ensino Superior (CAPES) scholarship to VM."

  "A.H. 

RGPIN-330527-20

Natural Sciences and Engineering Research Council of Canada (NSERC)

https://www.nserc-crsng.gc.ca

5. We note that Figures 1,2 and 4 in your submission contain map/satellite images which may be copyrighted. All PLOS content is published under the Creative Commons Attribution License (CC BY 4.0), which means that the manuscript, images, and Supporting Information files will be freely available online, and any third party is permitted to access, download, copy, distribute, and use these materials in any way, even commercially, with proper attribution. For these reasons, we cannot publish previously copyrighted maps or satellite images created using proprietary data, such as Google software (Google Maps, Street View, and Earth). For more information, see our copyright guidelines: http://journals.plos.org/plosone/s/licenses-and-copyright.

a. You may seek permission from the original copyright holder of Figures 1,2 and 4 to publish the content specifically under the CC BY 4.0 license.  

Additional Editor Comments:

The reviewers point out technical shortcomings, particularly in the selection of precipitation datasets and in the statistical analysis on the significance on difference in their association with vegetation data and their robustness to, for example, the choice of metric (e.g. R2 vs. rank correlation). These should be addressed in a revision.

Reviewers' comments:

Reviewer's Responses to Questions

**Comments to the Author**

1. Is the manuscript technically sound, and do the data support the conclusions?

Reviewer #1: Partly

Reviewer #2: Partly

2. Has the statistical analysis been performed appropriately and rigorously? 

Reviewer #1: Yes

Reviewer #2: Yes

3. Have the authors made all data underlying the findings in their manuscript fully available?

Reviewer #1: Yes

Reviewer #2: No

4. Is the manuscript presented in an intelligible fashion and written in standard English?

Reviewer #1: Yes

Reviewer #2: Yes

5. Review Comments to the Author

Reviewer #1: This manuscript evaluates a marginally novel idea (i.e., the use of EVI and tree rings to validate global precipitation datasets). But in the end there's just not much new here. For example, the results show that the products agree in areas with good data coverage, which has been well established in the interpolation literature for a long time now. The most notable result otherwise is that UDEL-TS performs better prior to the 1930s (mostly in North Asia and the Himalayas) according to the tree-ring benchmark, which is a feel-good moment for the Blue Hens out there but not really a major finding for climate science in general. Finally, there's no particular rationale for the satellite and reanalysis products used in the study; prominent satellite product omissions include IMERGEHHE, CHIRPS, GPCP, and MSWEP, and important reanalysis omissions include MERRA-2, CFSRv2, and JRA-55. Some of these omissions are known to outperform some of the products included in the manuscript.

Reviewer #2: Review of ‘Validation of global precipitation time series products against tree ring records and

remotely sensed vegetation greenness’ by Manvailer and Hamann

This paper compares six different global gridded precipitation products. Performance is evaluated by examining yearly correlations (variance explained) against two biological indices - vegetation greenness and tree ring data.

I believe that this approach is novel and warrants publication. However, there are aspects of the paper that I believe need to be addressed prior to publication. The main issues are:

-I need to be convinced about the validity of many conclusions drawn by the authors. For example, the authors state that: 'The results show that for areas with high density of weather station coverage (e.g. the United States and Europe), all products provide near-identical estimates.' This statement is misleading since all products provide near-identical estimates for all zones, with no particular distinction for Europe and North America. Given the minimal differences (with the exception of PERSIANN), there is a need for statistical testing to determine if any of the differences are actually statistically significant. A Kolmogorov-Smirnov (KS) distribution test prior to averaging the data could be an option, but I'm sure there are others. Ultimately, one could argue that the main conclusions of this study should be: 'All products behave essentially the same over most of the globe, with the exception of PERSIANN.'

-The claim that UDEL-TS was superior to all other products for long-term records is somewhat misleading. It would be more accurate to clearly state that it was better than only two other products and was not superior for analyzing long-term time series, but only prior to 1930. When considering the full length of the time series, one could argue that GPCC is the superior product.

-The paper does not discuss or mention the scales of the different datasets and the potential impact this might have on the results. Tree ring data, for example, is a very local product with a lot of spatial variability.

-The limitations of the study could be much better outlined. There are many possible measures of fitness when comparing products, and the one used in this study (correlation) is useful but gives a rather incomplete view. A highly correlated product could still have a large bias and high RMSE, thereby strongly limiting its usefulness. The authors acknowledge that they cannot evaluate bias, but do so without any further discussion. In my field, a product with a large bias would be automatically discarded. Finally, while monthly average values are useful, daily data is also critical for many applications, making sub-monthly variability important.

I am not sure why the authors were surprised by the PERSIANN results. There are many studies that have shown (either by comparison against stations, or using hydrological models) that PERSIANN is not as good as most other available products.

L10 ‘are’ instead of ‘is’

L42 challenging and not challanging

L101 Why were those 6 products chosen and not any others. For example, Beck et al, 2017 did a comparison of 22 precipitation datasets. Beck, H. E., Vergopolan, N., Pan, M., Levizzani, V., Van Dijk, A. I., Weedon, G. P., ... & Wood, E. F. (2017). Global-scale evaluation of 22 precipitation datasets using gauge observations and hydrological modeling. Hydrology and Earth System Sciences, 21(12), 6201-6217.

L!35 I looked very hard and could not find Table 1. I believe it has not been submitted.

L151-152 why was long and medium frequency filtered out? These are important characteristics of precipitation

L194 not clear what is meant by ‘dark gray coverage’. And I really studied the Figure.

L204-205 Not entirely clear as to what was used for the variance estimation. What is the cumulative annual EVI area under the curve ? If I get this right, a single value at 8 months is compared to the cumulative precipitation for the first 8 months for each year ? Please clarify.

Conclusion is essentially a copy of parts of the abstract, I believe the space could be used much more efficiently.

6. PLOS authors have the option to publish the peer review history of their article (what does this mean?). If published, this will include your full peer review and any attached files.

Reviewer #1: No

Reviewer #2: No

---

## [Author Response · Author response to Decision Letter 0]

5 Jan 2024

Thank you for the opportunity to revise this manuscript. One suggestion made by both reviewers was to include another high-quality remote-sensing based product expanded our analysis. The recommended dataset (MSWEP) performed very well, with improvements especially pronounced where weather station coverage is generally low (e.g. Africa). As directed in the Decision Letter our response to specific comments is provided in a separate file submitted here. We look forward to hearing from you.

---

## [Editor Report · Decision Letter 1]

6 Feb 2024

Validation of global precipitation time series products against tree ring records and remotely sensed vegetation greenness

PONE-D-23-14494R1

Dear Dr. Manvailer,

We’re pleased to inform you that your manuscript has been judged scientifically suitable for publication and will be formally accepted for publication once it meets all outstanding technical requirements.

Kind regards,

Nir Y. Krakauer

Academic Editor

PLOS ONE

Additional Editor Comments (optional):

Figure 3 is not legible in the black-and-white format provided. A color version should be used and checked for clarity and completeness.
---

## [Editor Report · Acceptance letter]

20 Feb 2024

PONE-D-23-14494R1 

PLOS ONE

Dear Dr. Manvailer, 

I'm pleased to inform you that your manuscript has been deemed suitable for publication in PLOS ONE. Congratulations! Your manuscript is now being handed over to our production team.

Kind regards, 

on behalf of

Dr. Nir Y. Krakauer 

Academic Editor

PLOS ONE